# Prevalence and risk factors for wheeze, decreased forced expiratory volume in 1 s and bronchoconstriction in young children living in Havana, Cuba: a population-based cohort study

Ramón Suárez-Medina,[1] Silvia Venero-Fernández,[1] Vilma Alvarez-Valdés,[2] Nieves Sardiñas-Baez,[2] Carmona Cristina,[2] Maria Loinaz-Gonzalez,[2] Zunilda Verdecia-Pérez,[2] Barbara Corona-Tamayo,[2] Maria Betancourt-López,[1] John Britton,[3] Andrew W Fogarty [4]

For numbered affiliations see end of article.

**Correspondence to**
Dr Ramón Suárez-Medina; ramonsm@infomed.sld.cu

## ABSTRACT

**Objectives** Asthma has not been extensively studied in low-income and middle-income countries, where risk factors and access to treatment may differ from more affluent countries. We aimed to identify the prevalence of asthma and local risk factors in Havana, Cuba.

**Setting** Four municipalities in Havana, Cuba.

**Participants** A population-based cohort study design of young children living in Havana, Cuba. Children were recruited from primary care centres at age 12–15 months.

**Primary and secondary outcome measures** Data on wheeze in the past 12 months, asthma treatment and environmental exposures collected regularly until the age of 6 years, when forced expiratory volume in 1 s ($FEV_1$) and reversibility to aerosolised salbutamol were also measured.

**Results** 1106 children provided data at the age of 6 years old. The prevalence of wheeze in the previous 12 months was 422 (38%), and 294 (33%) of the study population had bronchodilatation of 12% or more in $FEV_1$ after administration of inhaled salbutamol. In the previous 12 months, 182 (16%) of the children had received inhaled corticosteroids, 416 (38%) salbutamol inhalers and 283 (26%) a course of systemic steroids.

Wheeze in the first year and a family history of asthma were both positively associated with bronchodilatation to inhaled salbutamol (1.94%; 95% CI 0.81 to 3.08 and 1.85%; CI 0.14 to 3.57, respectively), while paracetamol use in the first year was associated with wheeze at 6 years (OR 1.64, 95% CI 1.14 to 2.35). There were large differences in $FEV_1$, bronchodilatation and risk of wheeze across different geographical areas.

**Conclusions** Asthma is common in young children living in Havana, and the high prevalence of systemic steroids administrated is likely to reflect the underuse of regular inhaled corticosteroids. If replicated in other comparable low-income and middle-income countries, this represents an important global public health issue.

## INTRODUCTION

Asthma is a global disease that affects approximately 11% of 6-year-old children,[1] but with

## Strengths and limitations of this study

► There have been many epidemiological studies of asthma in children who live in developed countries, but few population-based studies that have used objective measures of asthma in low-income and middle-income countries.

► The prevalence of bronchoconstriction in young children living in Havana was measured using both subjective and objective measures of asthma.

► Life course exposure data were available from birth onwards.

► Data on lung development were objectively measured using spirometry.

► These data are from one low-income and middle-income countries and thus not generalisable to other nations with different economies and healthcare systems.

marked regional differences in prevalence.[1] The aetiology of asthma is complex, and involves a range of environmental exposures that are likely to have differential impacts at different ages over the human lifetime.[2–7] The early years of childhood is a particularly important period as this is when the lungs and immune system are developing rapidly, and lung function in children is a key determinant of health in adulthood.[8]

This study was established as a consequence of concerns among public health specialists and clinicians that asthma was becoming a large problem in Cuba. It aimed to determine the prevalence of wheeze in young children living in Cuba, and to identify modifiable risk factors for wheezing, reduced lung function and reversible bronchoconstriction. The role of infection in early life in the development

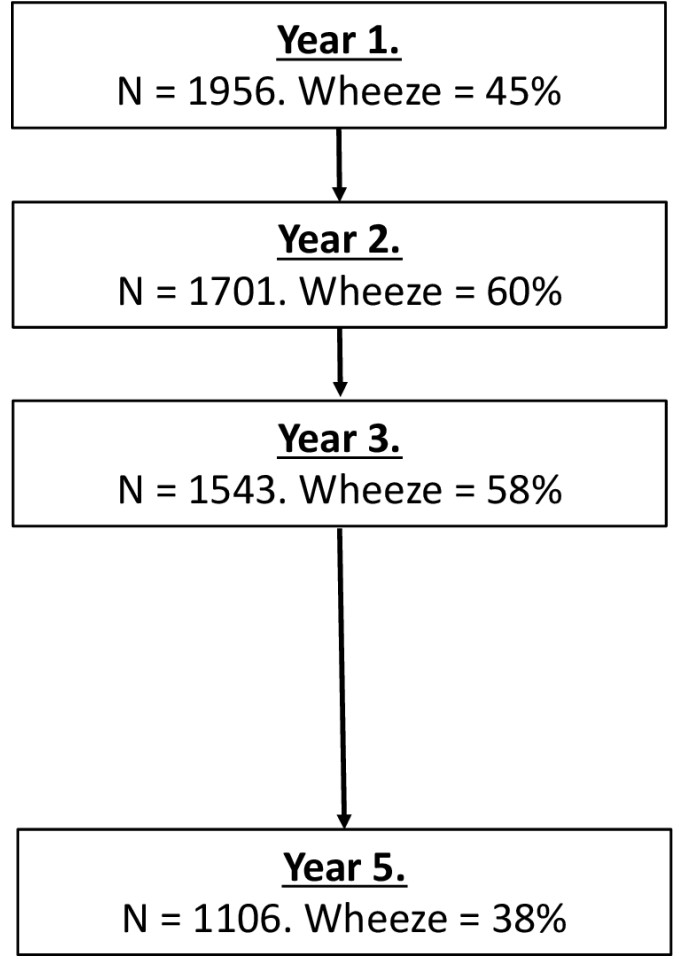

**Year 1.**
N = 1956. Wheeze = 45%

**Year 2.**
N = 1701. Wheeze = 60%

**Year 3.**
N = 1543. Wheeze = 58%

**Year 5.**
N = 1106. Wheeze = 38%

**Figure 1** Flow diagram of participants and data collection.

of allergic disease remains unclear.[9] The main hypothesis of interest was that infection with parasites,[10] *Helicobacter pylori*,[11] dengue[12] or systemic inflammation[13] may be associated with wheeze or bronchoconstriction. Exposure to environmental tobacco smoke and paracetamol had previously been observed to be positively associated with wheeze[14] or atopic dermatitis[15] symptoms, respectively, and so the association of these exposures with wheeze and bronchoconstriction were also studied. Finally, as growth from in utero onwards may also be related to development of asthma and growth of the lungs,[16] anthropometric measures from birth onwards were also considered. The study design is a prospective population-based study of an existing cohort of children followed from approximately 1 year of age for 5 years.

## METHODS
### Study population and sample collection
The study population is a cohort of 1956 children aged 12–15 months who were randomly selected from the general population from four municipalities across Havana in 2010 and 2011.[14 15 17] The response rate of those who were eligible to participate initially was 96%.[14] Data were collected by a standardised questionnaire that was administered by a member of the study team at baseline and subsequently at 2 years, 3 years and 5 years later. This included a number of health and lifestyle questions that were answered by the parent or guardian and particular attention was paid to parental/guardian reported wheeze in the past 12 months using the methodology developed for the ISAAC epidemiological studies of asthma,[18] use of asthma medication in the past 12 months and exposure to environmental tobacco smoke. The child's weight, height and mid-arm circumference in both arms were collected at each study visit. Historical baseline data including birth weight and height were collected from the primary care centre records. At each annual follow-up study the participants' guardians were asked if the child had received a medical diagnosis of dengue infection in the previous year (in Cuba all positive diagnoses of dengue infection are clinically confirmed using a serological IgM assay) and a blood sample was collected from children to measure circulating eosinophil levels. This sample was stored at −20°C and subsequently defrosted and analysed for dengue IgG serology to generate an antibody index,[19] serum IgE,[17] highly sensitive C-reactive protein (hsCRP, SpinReact, Spain),[20] toxoplasmosis IgG antibodies[21] and toxocariasis IgG antibodies (DRG Instruments, Germany). A faecal sample was also collected at each review and stored at −20°C, and later examined for *H. pylori* using the faecal antigen test (SpinReact, Spain) and intestinal parasites using the Kato-Katz test (Campiñas Medical COMI, Brazil).

### Lung function
Forced expiratory volume in 1 s ($FEV_1$) and forced vital capacity were measured in accordance with American Thoracic Society/European Respiratory Society criteria[22] using spirometers (CareFusion Micro I) calibrated each day to allow for local climatic change. The best value of $FEV_1$ within a threshold of repeatability of 200 mL was used as the final value. Aerosolised salbutamol (300 μg) was then administered via a spacer and after 15-min lung function was measured again to quantify airway reversibility. In children who provided a postbronchodilator $FEV_1$ that was less than the baseline value, they were considered as having no reversibility to bronchodilator as this was likely to be due to fatigue.

### Allergen skin prick testing
Skin prick testing was used to determine allergy to mite, cat, grass, cockroach, fungus, mosquitos, wheat and soy (allergens from Diater, Argentina except mite allergen from Biocen, Cuba). For each test a drop of allergen solution was placed on the skin and a lancet used to break the skin. After 15 min, the skin weal was measured at its maximum diameter, and also perpendicularly, and a mean value generated. The final skin prick test result was calculated by subtracting the saline result from the allergen. A value of ≥3 mm was used to define a positive atopic result for each allergen, and atopy was defined as any positive skin prick test.

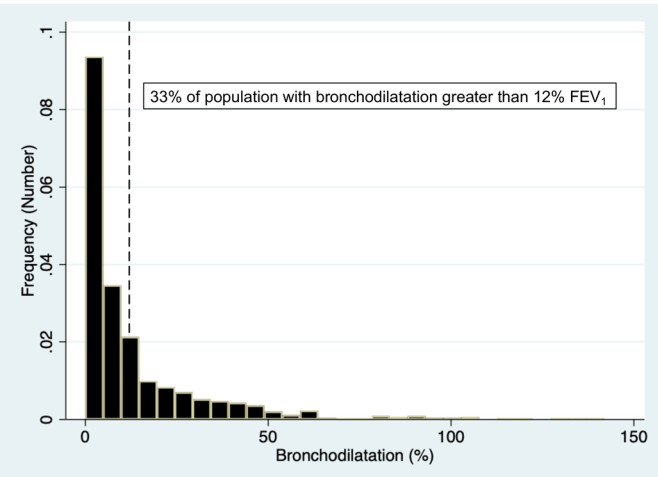

**Figure 2** Histogram of per cent increase in forced expiratory volume in 1 s (FEV$_1$) in study population (n=903 children).

## Statistical analysis

The main outcome variables were FEV$_1$, per cent increase in FEV$_1$ after to inhaled salbutamol and wheeze in the past 12 months. The main exposure variables were grouped into three categories:

1. Prior exposures: wheeze in the first year of life, family history of asthma, nursery attendance, birth weight, birth height, duration of breastfeeding, blood IgE and eosinophils at 1 year old; faecal *H. pylori* antigen at 2 and 3 years old; blood hsCRP, dengue IgG serology, eosinophils, toxoplasmosis serology, IgE at 3 years old and any prior medical diagnosis of dengue infection.
2. Cross-section exposures: number of smokers living in the home, current weight, current height, mean arm circumference, municipality of residence.
3. Biomarkers of current infection and inflammation: *H. pylori* stool antigen, toxoplasmosis IgG serology, dengue IgG serology, blood hsCRP, eosinophils, IgE, toxocariasis serology and atopy. Less than 2% of children has current gastro-intestinal parasite infection and these data were not analysed further.

Statistical analysis used linear and logistic regression adjusting for sex and age in months as a priori confounding factors, and also adjusted for clustering by municipality of residence. As height was associated with FEV$_1$, all analyses of this outcome measure also adjusted for height to ensure that the analyses were not confounded by somatic growth. $\chi^2$ tests were used to explore differences in categorical exposures for binary outcome measures. All analyses used Stata V.14 statistical software.

## Patient and public involvement

The study was designed as a consequence of concerns from the Cuban public health and clinical communities about asthma morbidity. The patients were not involved in the design of the study and patients did not receive a copy of the results. We thank the patients and their families for their participation.

Consent on the behalf of the child was provided by the attending parent or guardian.

## RESULTS

Data were available for 1106 children, of whom 422 (38%) had reported wheeze in the previous year (figure 1). Wheeze in the first year of life was reported in 514 (46%) current participants, while there was a prevalence of wheeze in the first year of life of 42% (358 children) for those who did not participate in the study at the age of 6 years (p=0.055). Nine hundred and thirteen (83%) children provided lung function data, and of these 903 (99%) children had their reversibility to salbutamol measured. The mean FEV$_1$ was 1.13 L (SD 0.31), and 294 (33%) had an increase in FEV$_1$ of more than 12% after administration of aerosolised salbutamol (figure 2). Two hundred and eighty-three (26%) of the current study population were reported to have received systemic steroids in the previous 12 months (table 1).

### Risk factors for wheeze in the past 12 months at 6 years old

The analysis of the association of exposures with wheeze in the past 12 months is presented in table 2. Both wheeze (OR 1.89; 95% CI 1.65 to 2.16) and paracetamol use (OR 1.64; 95% CI 1.14 to 2.35) in the first year of life along with a family history of asthma (OR 1.66; 95% CI 1.40 to 1.97) were associated with wheeze at 6 years. A positive *H. pylori* faecal antigen test at age 2 years was negatively associated with wheeze in the past 12 months (OR 0.57; 95% CI 0.40 to 0.82), but this association was not observed for *H. pylori* at the age of 3 years or 6 years. The number of smokers in the household was a strong risk factor for wheeze in the past 12 months (p<0.001 for trend), with homes with two or more smokers having an OR of the child having wheeze in the past 12 months of 2.08 (95% CI 1.71 to 2.54) compared with those homes with no smokers. The municipality of residence was associated with wheeze in the past 12 months (p=0.04, $\chi^2$ test), with children living in Cerro municipality having the highest risk of wheeze (OR 1.72 compared with Arroyo Naranjo; 95% CI 1.61 to 1.84). These differences were not substantially modified by adjusting for the number of smokers in the home.

### Risk factors for decreased FEV$_1$ at 6 years old

The analysis of the association of a number of exposures with FEV$_1$ is presented in table 3. A number of measures of somatic growth were positively associated with FEV$_1$ at 6 years. These were birth height (14 mL /cm height at birth; 95% CI 6 to 23), current height (11 mL /cm; 95% CI 5 to 18), current weight (11 mL /kg; 95% CI 3 to 18) and current mean arm circumference (12 mL /cm; 95% CI 1 to 24). The number of smokers living in the home was not associated with lung function, but the municipality of residence was strongly associated with current FEV$_1$ (p<0.001, Ananlysis of Variance/ANOVA test), with children living in La Lisa having the lowest lung function (−95 mL compared with Arroyo Naranjo, 95% CI −126 to

**Table 1** Description of study population

| | Total (n=1106) | Provided FEV$_1$ (n=913) |
|---|---|---|
| Male sex (%) | 575 (52) | 473 (52) |
| Mean age, months (range) | 74 (63– 83) | 74 (64 –83) |
| Mean FEV$_1$, L, (SD) | – | 1.13 (0.31) |
| Mean bronchodilation after salbutamol, % (SD) | – | 13.4 (20.1) n=903 |
| Wheeze in the past 12 months (%) | 422 (38) | 356 (39) |
| **Received inhaled steroids in the 12 months before (%)** | | |
| Year 1 | 89 (8) | 68 (7) |
| Year 2 | 176 (17) n=1051 | 153 (18) n=869 |
| Year 3 | 203 (18) | 163 (18) |
| Year 4 | – | – |
| Year 5 | 182 (16) | 150 (16) |
| **Received salbutamol inhaler in the previous 12 months (%)** | | |
| Year 5 | 416 (38) | 345 (38) |
| **Received intravenous or oral steroids in the 12 months before (%)** | | |
| Year 1 | 295 (27) | 244 (27) |
| Year 2 | 418 (40) n=1051 | 345 (40) n=869 |
| Year 3 | 407 (37) | 333 (36) |
| Year 4 | – | – |
| Year 5 | 283 (26) | 240 (26) |

FEV$_1$, forced expiratory volume in 1 s.

–64). These differences were not substantially modified by adjusting for the number of smokers in the home. No measures of infection or inflammation were associated with lung function.

### Risk factors for bronchodilatation after inhaled salbutamol at 6 years old

The association of exposures with change in FEV$_1$ from baseline after aerosolised salbutamol was administered is presented in table 4. Any wheeze in the first year of life was positively associated with bronchodilatation (1.94%; 95% CI 0.81 to 3.08) as was a family history of asthma (1.85; 95% CI 0.14 to 3.57), but there was no relation with wheeze in the past 12 months (3.61; 95% CI –5.80 to 13.02). Children with a higher birth weight had a lower risk of bronchodilatation (–2.67%; 95% CI –4.49 to –0.84). IgE at the age of 1 year was positively associated

with a higher risk of bronchodilatation (1.68%; 95% CI 0.54 to 2.82), but not IgE at age of 3 years or 6 years.

The numbers of smokers living in the child's home was not associated with bronchodilation, but the municipality of residence was again a strong determinant of current bronchodilatation (p=0.002, ANOVA test), with children living in La Lisa having the highest increase in FEV$_1$ after administration of inhaled salbutamol at 6 years old (6.24% compared with Arroyo Naranjo; 95% CI 5.56 to 6.91). These differences were not substantially modified by adjusting for the number of smokers in the home.

### DISCUSSION

The most striking observation from our cohort study of young children living in Havana is that the prevalence of wheeze in these children was high, with 38% reporting wheeze in the past 12 months. Similarly, there was a high prevalence of bronchoconstriction using the objective measure of lung function reversibility to inhaled salbutamol, with 33% of the study population having an increase in FEV$_1$ of 12% or more. Over a quarter of these 6 years old children have received systemic steroids in the previous 12 months, suggesting that asthma control was suboptimal. The main consistent association is that municipality of residence is a strong risk factor for wheeze, low lung function and bronchoconstriction. A history of wheeze in early life, exposure to paracetamol in the first year of life and current environmental tobacco smoke were risk factors for wheeze, anthropometric measures were positively associated with higher FEV$_1$, and wheeze in the first year of life and lower birth weight were associated with untreated bronchoconstriction.

### Strengths and limitations of the data

This is the first cohort study of young children living in Cuba that has collected extensive life course data to evaluate risk factors for asthma using objective measures of lung function and reversibility along with laboratory measures of exposure to infection. The strengths of our data include the prospective nature of the data collected with exposures and outcome measures that were selected to permit exploration of risk factors for asthma specifically within an urban Cuban environment, where the lifestyle, environment and microbial exposures are very different to more temperate, developed countries. The measurement of lung function is a challenge in young children, yet measurements were obtained in 82% of eligible children. The availability of lung function measurements along with reversibility to inhaled salbutamol is a particular strength of these data, as it provides an objective measure that may reflect different aspects of lung health compared with self-reported symptoms.

Our dataset does have some limitations. We initially recruited 1956 children to the cohort at the age of 1 year with a response rate of 96% eligible children, and approximately 5 years later were able to collect data on 1106 (57%) of these. This was mainly a consequence of

**Table 2** Association of exposures with wheeze in past 12 months

| Total number=1106 | No (%, SD) | OR of wheeze (95% CI) |
|---|---|---|
| **Prior exposures** | | |
| **Any wheeze in first year of life** | 514 (46) | **1.89 (1.65 to 2.16)** |
| **Family history of asthma** | 614 (56) | **1.66 (1.40 to 1.97)** |
| Attendance at nursery | 159 (14) | 0.84 (0.57 to 1.24) |
| **Paracetamol in first year of life** | 256 (23) | **1.64 (1.14 to 2.35)** |
| Mean birth weight, N, kg (SD) n=1104 | 3.31 (0.51) | 0.87 (0.75 to 1.01) |
| Mean birth height, cm (SD) | 50.2 (2.4) | 0.95 (0.89 to 1.00) |
| Breast feeding ≥4 months | 608 (55) | 0.83 (0.66 to 1.03) |
| *Age=1 year* | | |
| Mean log IgE, n=885 | 3.38 (1.47) | 1.06 (0.88 to 1.27) |
| Mean log eosinophils, n=856 | −2.11 (1.05) | 0.93 (0.77 to 1.13) |
| *Age=2 years* | | |
| **Helicobacter stool +ve, n=1067** | 40 (4) | **0.57 (0.40 to 0.82)** |
| *Age=3 years* | | |
| Mean log CRP, SD, n=986 | −2.07 (2.70) | 1.01 (0.95 to 1.07) |
| Log dengue IgG, n=865 | −0.44 (2.00) | 0.99 (0.96 to 1.02) |
| Helicobacter stool +ve, n=951 | 58 (6) | 1.01 (0.64 to 1.60) |
| **Mean log eosinophils, n=1039** | −1.77 (1.23) | **0.91 (0.86 to 0.96)** |
| Toxoplasmosis serology +ve, n=966 | 565 (58) | 1.15 (0.82 to 1.64) |
| Mean log IgE, n=986 | 3.70 (1.53) | 1.14 (0.98 to 1.31) |
| Medical diagnosis of dengue infection | 93 (8) | 1.40 (0.91 to 2.14) |
| **Current exposures** | | |
| **Male sex** | 575 (52) | **1.35 (1.00 to 1.82)** |
| **Age (months), (range)** | 74 (63 to 83) | **0.97 (0.96 to 0.99)** |
| Mean current weight, kg, n=1062 | 23.3 (11 to 51) | 1.01 (0.97 to 1.06) |
| Current height, cm, n=1057 | 119 (7) | 1.00 (0.97 to 1.03) |
| Mean arm circumference, cm, n=1062 | 18.3 (2.4) | 1.02 (0.95 to 1.08) |
| *Number of current smokers in house* | | |
| *0* | 546 (49) | 0 |
| *1* | 322 (29) | **1.61 (1.22 to 2.12)** |
| *≥2* | 238 (22) | **2.08 (1.71 to 2.54)** |
| *Municipality of residence* | | $P_{TREND}$ <0.001 |
| **Arroyo Naranjo** | 455 (41) | **0** |
| **Cerro** | 139 (13) | **1.72 (1.61 to 1.84)** |
| **Habana del Este** | 307 (28) | **0.86 (0.84 to 0.88)** |
| **La Lisa** | 205 (19) | **1.24 (1.21 to 1.27)** |
| | | **P=0.04** |
| **Current infection and blood assays** | | |
| Helicobacter stool antigen +ve, n=756 | 11 (1) | 1.45 (0.63 to 3.33) |
| Toxoplasmosis +ve, n=759 | 487 (64) | 0.97 (0.70 to 1.34) |
| Mean log dengue IgG serology, n=758 | 1.67 (1.53) | 0.95 (0.90 to 1.01) |
| Log CRP, n=757 | −0.83 (1.41) | 0.99 (0.86 to 1.15) |
| Mean log eosinophils, n=868 | −1.62 (1.45) | 1.02 (0.90 to 1.15) |
| **Mean log IgE, n=759** | 4.51 (0.97) | **1.27 (1.15 to 1.41)** |
| Toxocariasis +ve, n=673 | 64 (9) | 0.96 (0.70 to 1.33) |
| **Any atopy* (n=857)** | 52 (6) | 0.81 (0.43 to 1.53) |

Results in bold font have a probability of <0.05.
CRP, C-reactive protein.

| Table 3 Association of exposures with FEV₁ | | |
|---|---|---|
| **Total number=903** | **No (%, SD)** | **FEV₁, mL (95% CI)** |
| **Prior exposures** | | |
| Any wheeze in first year of life | 416 (46) | −16 (−50 to 18) |
| Family history of asthma | 510 (56) | −23 (−89 to 43) |
| Attendance at nursery | 141 (16) | −48 (−149 to 53) |
| Paracetamol in first year of life | 215 (24) | 8 (−36 to 51) |
| Mean birth weight, N, kg (SD) n=901 | 3.32 (0.53) | 43 (−19 to 104) |
| **Mean birth height, cm (SD)** | **50 (2)** | **11 (5 to 18)** |
| Breast feeding ≥4 months | 503 (56) | 18 (−65 to 101) |
| *Age=1 year* | | |
| Mean log IgE, n=456 | 3.44 (1.39) | −19 (−50 to 12) |
| Mean log eosinophils, n=451 | −2.14 (1.04) | −13 (−56 to 30) |
| *Age=2 years* | | |
| Helicobacter stool +ve, n=589 | 29 (5) | −78 (−270 to 115) |
| *Age=3 years* | | |
| Mean log CRP, SD, n=614 | −2.16 (2.71) | 2 (−9 to 12) |
| Log dengue IgG, n=545 | −0.43 (2.00) | 7 (−10 to 25) |
| Helicobacter stool +ve, n=775 | 51 (7) | −39 (−132 to 54) |
| Mean log eosinophils, n=652 | −1.78 (1.26) | 22 (−5 to 49) |
| Toxoplasmosis serology +ve, n=606 | 362 (60) | 48 (−46 to 143) |
| Mean log IgE, n=614 | 3.68 (1.57) | 8 (−21 to 37) |
| Medical diagnosis of dengue infection | 78 (9) | −27 (−59 to 5) |
| **Current exposures** | | |
| Male sex | 468 (52) | 36 (−23 to 96) |
| Mean age (months), (range) | 74 (64 −83) | 4 (−12 to 21) |
| Wheeze in past 12 months | 353 (39) | −62 (−160 to 35) |
| **Mean weight, kg, n=902 (range)** | **23.5 (11 −51)** | **11 (3 to 18)** |
| **Current height, cm, n=903 (range)** | **119 (90 −175)** | **8 (2 to 14)** |
| **Mean arm circumference, cm n=901** | **18.4 (2.4)** | **12 (1 to 24)** |
| *Number of current smokers in house* | | |
| *0* | 447 (49) | 0 |
| *1* | 261 (29) | 0 (−20 to 19) |
| *≥2* | 195 (22) | −39 (−108 to 30) |
| *Municipality of residence* | | |
| **Arroyo Naranjo** | **371 (41)** | **0** |
| **Cerro** | **119 (13)** | **74 (31 to 117)** |
| **Habana del Este** | **247 (27)** | **048 (39 to 57)** |
| **La Lisa** | **166 (18)** | **−95 (−126 to −64)** |
| | | **P<0.001** |
| **Current infection and blood assays** | | |
| Helicobacter stool antigen +ve, n=685 | 10 (1) | −10 (−228 to 207) |
| Toxoplasmosis +ve, n=688 | 437 (64) | 8 (−49 to 65) |
| Mean log dengue IgG serology, n=687 | 1.63 (1.56) | −5 (−24 to 14) |
| Log CRP, n=686 | −0.87 (1.38) | −9 (−37 to 20) |
| Mean log eosinophils, n=785 | −1.61 (1.47) | −5 (−41 to 31) |
| Mean log IgE, n=688 | 4.52 (0.97) | −12 (−87 to 63) |

Continued

**Table 3** Continued

| Total number=903 | No (%, SD) | FEV$_1$, mL (95% CI) |
|---|---|---|
| Toxocariasis +ve, n=667 | 57 (9) | −15 (−125 to 95) |
| **Any atopy* (n=818)** | 52 (6) | −54 (−182 to 74) adjusted for sex, age in months, current height and clustering by municipality |

Results in bold font have a probability of <0.05.
* Any allergen skin prick test >3 mm larger than saline control.
CRP, C-reactive protein; FEV$_1$, forced expiratory volume in 1 s.

the recent changes in Cuba that have made it easier for the population more mobile, and hence many children left the policlinic from where they were initially recruited when their family moved their residence. There was only a small difference in the prevalence of wheeze in the first year of life between those who provided data at the age of 6 years (46%) and those who did not (42%), so loss to follow-up is unlikely to constitute a major source of bias. Collecting data on parental or self-reported wheeze using questionnaires is challenging, and although we used the optimal methodology from the ISAAC international studies of allergic disease[18] it is possible that some of the cases of wheeze were not asthma. Similarly, parental reported asthma medication use may be prone to recall bias, although as acute asthma in children is an upsetting family event it is likely to be remembered accurately although the time period possibly less so. This included a number of health and lifestyle questions that were answered by the parent or guardian and particular attention was paid to parental/guardian reported wheeze in the past 12 months using the methodology developed for the ISAAC epidemiological studies of asthma,[18] use of asthma medication in the past 12 months and exposure to environmental tobacco smoke. There are no validated lung function normal values in Cuban children aged 6 years old and as a consequence we were unable to generate reliable per cent predicted values of the lung function in our study population. Finally, we used the 200 mL as the definition of repeatability of FEV$_1$, which is a value generally used in adults. As a consequence, our population provided FEV$_1$ measures that will have a higher measurement error than observed in clinical patients. Nonetheless, these data of the peak FEV$_1$ value records have an epidemiological value as the methodology of data collection was standardised across the whole population. This is supported by the observation of the association between FEV$_1$ and height, which are both different aspects of somatic growth.

Like many cohorts, our study began as a cross-sectional population-based that aimed to explore causes of the high prevalence of asthma in Habana, Cuba. Further funding permitted this to become a prospective study, with a particular emphasis on infections found in the tropics. As a consequence, we have tested a variety of analyses and present the results in their entirety. While we recognise that it is possible to generate a number of papers analysing different hypotheses from these data, this approach has the advantage of presenting a complete body of work that hopefully can inform researchers who are interested in this area. While we have considered that sex and age in months are a priori confounding factors and adjusted for them in all analyses, we have refrained for searching through all the data to look for other possible confounding factors and possible interactions. The one exception to this is to explore the hypothesis that cigarette smoking may explain the differences observed between municipalities for wheeze, FEV$_1$ and bronchodilatation as this is an obvious question that was driven by Cuban public health concerns about the hazards of exposure to secondhand tobacco smoke.

### Risk factors for wheeze in the previous 12 months at 6 years old

Wheeze in the first year of life is associated with wheeze in later childhood,[23] and may reflect the establishment of the asthmatic phenotype in susceptible children. The observation that paracetamol consumption in early life is associated with wheeze in later life[24] has been noted before, and our data support these observations. However, they do not demonstrate a causality, as an alternative explanation is reverse causality with the administration of more paracetamol to children who are more susceptible to infections and wheeze.[25]

The observation that *H. pylori* at the age of 2 years, but not 3 years old or 6 years old is associated with lower risk of wheeze at 6 years old is an interesting observation, that would be consistent with the hypothesis that age of exposure to infection is important. Data from other countries has demonstrated the *Helicobacter* is associated with lower rates of allergic disease but not wheeze in children,[11 26] but not in adults.[27] It is striking however that our prevalence of *H. pylori* infection was very low at less than 7% in the first 6 years of life, suggesting that this may be less important in Cuba compared with other countries.[11]

From a public health perspective, exposure to cigarette smoking and the municipality of residence are the most important exposures for wheeze at the age of 6 years. Secondhand cigarette smoke exposure is well recognised as a risk factor for wheeze in children,[5] and it is a concern that 51% of the homes in our study population contained

| Table 4 | Association of exposures with bronchodilatation after inhaled salbutamol | |
|---|---|---|
| **Total number=903** | **No (%, SD)** | **% change in FEV$_1$ (95% CI)** |
| **Prior exposures** | | |
| **Any wheeze in first year of life** | 417 (46) | **1.94 (0.81 to 3.08)** |
| **Family history of asthma** | 510 (56) | **1.85 (0.14 to 3.57)** |
| Attendance at nursery | 138 (15) | −3.21 (−9.10 to 2.68) |
| Paracetamol in first year of life | 213 (24) | −0.67 (−4.68 to 3.35) |
| Mean birth weight, kg (SD) n=901 | 3.32 (0.53) | **−2.67 (−4.49 to −0.84)** |
| Mean birth height, cm | 50 (2) | 0.06 (−0.56 to 0.69) |
| Breast feeding ≥4 months | 503 (56) | 0.61 (−2.87 to 4.09) |
| *Age=1 year* | | |
| **Mean log IgE, n=455** | 3.42 (1.39) | **1.68 (0.54 to 2.82)** |
| Mean log eosinophils, n=449 | −2.13 (1.04) | −0.16 (−2.54 to 2.21) |
| *Age=2 years* | | |
| Helicobacter stool +ve, n=586 | 28 (5) | −6.41 (−14.94 to 2.10) |
| *Age=3 years* | | |
| Mean log CRP, SD, n=615 | −2.12 (2.71) | −0.28 (−0.88 to 0.32) |
| Log dengue IgG, n=550 | −0.43 (2.01) | −0.38 (−1.50 to 0.74) |
| Helicobacter stool +ve, n=776 | 51 (6) | −1.40 (−7.30 to 4.51) |
| Mean log eosinophils, n=653 | −1.78 (1.26) | −0.39 (−4.45 to 3.68) |
| Toxoplasmosis serology +ve, n=607 | 362 (60) | −0.45 (−7.06 to 6.16) |
| Mean log IgE, n=615 | 3.68 (1.57) | 0.23 (−1.81 to 2.27) |
| Medical diagnosis of dengue infection | 78 (9) | −0.58 (−8.55 to 7.38) |
| **Current exposures** | | |
| Male sex | 466 (52) | 2.22 (−0.28 to 4.71) |
| Age (months), (range) | 74 (64– 83) | −0.13 (−0.47 to 0.20) |
| Wheeze in past 12 months | 353 (39) | 3.61 (−5.80 to 13.02) |
| Mean current weight, kg, n=893 (range) | 24 (11 –51) | −0.23 (−0.54 to 0.08) |
| Current height, cm, n=893 | 119 (90– 175) | −0.08 (−0.50 to 0.34) |
| Mean arm circumference, cm, n=893 | 18.4 (2.4) | −0.14 (−0.81 to 0.54) |
| *Number of current smokers in house* | | |
| *0* | 446 (49) | 0 |
| *1* | 263 (29) | −0.87 (−5.89 to 4.14) |
| *≥2* | 194 (21) | 2.93 (−3.80 to 9.65) |
| *Municipality of residence* | | |
| **Arroyo Naranjo** | 368 (41) | 0 |
| **Cerro** | 119 (13) | **−1.34 (−2.56 to −0.11)** |
| **Habana del Este** | 249 (28) | **−0.07 (−0.46 to 0.31)** |
| **La Lisa** | 167 (18) | **6.24 (5.56 to 6.91)** |
| | | P=0.002 |
| **Current infection and blood assays** | | |
| Helicobacter stool antigen +ve, n=684 | 10 (1) | 8.89 (−9.17 to 26.96) |
| Toxoplasmosis +ve, n=687 | 438 (64) | −0.47 (−3.34 to 2.39) |
| Mean log dengue IgG serology, n=686 | 1.64 (1.56) | 0.30 (−1.44 to 2.03) |
| Log CRP, n=685 | −0.87 (1.40) | 0.37 (−1.64 to 2.38) |
| Mean log eosinophils, n=782 | −161 1.47) | 0.99 (−0.48 to 2.46) |
| Mean log IgE, n=686 | 4.52 (0.98) | 0.64 (−0.18 to 1.45) |

Continued

| Table 4 Continued | | |
|---|---|---|
| **Total number=903** | **No (%, SD)** | **% change in FEV$_1$ (95% CI)** |
| Toxocariasis +ve, n=666 | 58 (9) | −1.03 (−5.29 to 3.23) |
| **Any atopy* (n=816)** | **52 (6)** | **−2.75 (−10.15 to 4.60)** |

Adjusted for sex, age in months and clustering by municipality.
Results in bold font have a probability of <0.05.
*Any allergen skin prick test >3 mm larger than saline control.
CRP, C-reactive protein; FEV$_1$, forced expiratory volume in 1 s.

at least one smoker. The risk of wheeze was higher in the municipalities of Cerro and La Lisa, which suggest that there may be an environmental factor that predisposes to asthma symptoms such as higher levels of air pollution.[28–30]

### Risk factors for modified FEV$_1$ at 6 years old

The fact that children living in La Lisa municipality already have lower lung function at the age of 6 years is a major public health concern, and suggests that an exposure associated with living in this area may negatively impact on lung growth. La Lisa had by far the highest increase in FEV$_1$ after administration of aerosolised salbutamol, suggesting that bronchoconstriction is contributing to the low baseline FEV$_1$ and that a component of the apparent low lung function is reversible. La Lisa is located inland and while there is no obvious sources of excessive industrial pollution,[30] and we speculate that this may be associated with higher levels of air pollution as a consequence of the relative absence of sea winds compared with the other municipalities. Weight, height and mean arm circumference were all positively associated with FEV$_1$ at 6 years of age which is likely to be simply because they are all measures of somatic growth.

### Risk factors for bronchodilatation after inhaled salbutamol at 6 years old

Wheeze in the first year of life was associated with bronchodilatation after the administration of inhaled salbutamol at 6 years. This suggests that symptoms of asthma in early life are predictive of untreated bronchoconstriction 5 years later, and is consistent with the hypothesis that the asthmatic phenotype can be observed to persist from early life onwards.[23] The observation that birth weight is inversely associated with bronchoconstriction provides objective evidence to support the earlier epidemiological evidence that adults with higher birth weights have a lower risk of having a diagnosis of asthma.[16] The observation that a family history asthma is associated with bronchoconstriction also wheeze is consistent with the knowledge that there is a genetic component to asthma.[31]

### Self-reported wheeze and bronchoconstriction

One interesting observation from our data is that the wheeze in the past 12 months is not associated with either FEV$_1$ or the change in FEV$_1$ after administration of inhaled bronchodilator. Although it is possible that children who have had wheeze subsequently received asthma treatment, this lack of association between asthma symptoms and objective measures of asthma has been described previously.[32] This supports that concept that wheeze and reversibility of FEV$_1$ after administration of aerosolised salbutamol may measure different aspects of lung development and health.

### Public health implications of these data

The current study was designed in response to clinical concerns that asthma was becoming a public health problem in Havana. These data support that hypothesis, and in particular identify that living in certain municipalities may result in higher levels of currently unknown exposures that require public health intervention. It is possible that environmental air pollution may drive some of these geographical differences, and that studies of air pollution are urgently required in Havana. The prevalence of exposure to secondhand tobacco smoke in children living in Havana remains high, and this is an area where a combination of public health interventions including taxation, legislation that enforces bans of smoking in the workplace and public places, restrictions on tobacco product advertising and helping individual smokers quit smoking are known to be effective. We are exploring these hypotheses further by doing further objective measurements of particulate air pollution within the study population.

### Summary

Asthma is common in young children living in Havana, and the high prevalence of the use of systemic steroids probably reflects the underuse of regular inhaled corticosteroid prophylaxis treatments leading to the requirement for rescue treatment for wheezing. As societies urbanise, environmental air pollution may increase from a variety of sources. Cuba's economy has been inversely affected due to historical and political events,[33] and it remains under an economic embargo from the USA that has negatively impacted on healthcare.[34] This makes providing regular inhaled corticosteroids to all children who need them challenging. However, other low-income and middle-income countries also have difficult economic and environmental circumstances, and if these observations are replicated elsewhere, then this represents an important global public health issue.

**Author affiliations**
¹Instituto Nacional de Higiene, Epidemiología y Microbiología, La Habana, Cuba
²Dirección Municipal de Salud Pública municipios Cerro y Arroyo Naranjo, Habana, Cuba
³Division of Epidemiology and Public Health, UK Center for Tobacco and Alcohol Studies, University of Nottingham, Nottingham, UK
⁴NIHR Nottingham Biomedical Research Unit, Division of Epidemiology and Public Health, University of Nottingham, Nottingham, UK

**Acknowledgements**   We would like to thank the children and their parents and guardians for participating in this study. We would like to thank all of the staff at the municipalities and policlinics who helped in many ways with data collection and sample analysis.

**Contributors**   The study was conceived and designed by RS-M, SV-F, AWF and JB. Data were collected by RS-M, SV-F, VA-V, NS-B, CC, ML-G, ZV-P, BC-T and MBL. Data analysis was conducted by RS-M and AWF. RS-M, SV-F and AWF wrote the first draft of the manuscript. All authors critically reviewed the manuscript, provided intellectual content and approved the final version prior to submission for publication.

**Funding**   This study was funded by The Wellcome Trust (Grant Number 090375), The University of Nottingham, Nottingham University Hospitals Charity, NIHR Nottingham Biomedical Research Centre and a BMA Award (James Trust).

**Competing interests**   None declared.

**Patient consent for publication**   Not required.

**Ethics approval**   The study was approved by Ethics Committees in the Instituto Nacional de Higiene, Epidemiología y Microbiología, Cuba and at the University of Nottingham, UK.

**Provenance and peer review**   Not commissioned; externally peer reviewed.

**Data availability statement**   Data may be obtained from a third party and are not publicly available. Cuban regulations do not allow dissemination of national datasets. However, statistical analyses can be performed upon reasonable requests to the corresponding author.

**ORCID iD**
Andrew W Fogarty http://orcid.org/0000-0001-9426-977X

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
