## [Reviewer comments · BMJ Open]

ARTICLE DETAILS

TITLE (PROVISIONAL)	Prevalence and risk factors for wheeze, decreased forced expiratory volume in one second and bronchoconstriction in young children living in Havana, Cuba: A population-based cohort study
AUTHORS	Suarez-Medina, Ramon; Venero-Fernández, Silvia; Alvarez-Valdés, Vilma; Sardiñas-Baez, Nieves; Cristina, Carmona; Loinaz-Gonzalez, Maria; Verdecia-Pérez, Zunilda; Corona-Tamayo, Barbara; Betancourt-López, Maria; Britton, John; Fogarty, Andrew W

VERSION 1 – REVIEW

REVIEWER	Knut Øymar University of Bergen, Norway
REVIEW RETURNED	07-Oct-2019

GENERAL COMMENTS	Main comments  - This study seems to be asking a huge number of research questions which it is not primary designed for, no power analyses for this included, not a proper selection of variables and adjusted analyses. This is particularly for the high number of possible risk factors, of which the study provides no new information - I would suggest to concentrate on the main topic, the prevalence of symptoms, (possible lung function) and medication use. These are the main results of interest, providing that the authors can describe properly that the population is really unselected at the first inclusion. - Measuring and analyzing lung function as absolute volumes is not appropriate, the percent of predicted and the used reference material should be given - The authors justify that there is no selection bias from the first to the last follow up even though only 57% are followed the whole study. However, it must be better described the primary inclusion - are the included at health centers providing a selection of children with symptoms or randomly form all children in the municipality ? Could even though parents children with symptoms be more susceptible to accept ? Abstract: The primary selection procedure could be better described, it is not easy to understand how the participants are included at first even if it is stated that it is a population based study. Based on the results the population seem selected. The abstract includes several different issues, it is not a full congruence between aims, methods, results and conclusions. If fewer research questions are included in the manuscript, the abstract may be rewritten
---

	Methods:  - The inclusion of patients may be better described, even though more described in previous studies. As main results are different prevalences, it is crucial to understand if there has been any selection. Lung function; a threshold of 200 ml is applicable for adults, not for children. - At age six, a substantial proportion of children will not achieve acceptable spirometry data, here it seems that all children providing lung data are successful, making the results questionable. Even more, reversibility of 10% in this age as a measure of reversibility is low, and many children will probably achieve some improvement by chance or due to technical improvement after several measures. - A substantial numbers of unadjusted analyses (even though adjusted for age and sex) may not be appropriate. Fewer variables should have been carefully selected and included in multivariate analyses including sex and age, and considering interactions. - How are data regarding medications selected ? By questionnaire? A recall bias may be a problem, though perhaps than more probably an under-reporting - Ethics: were parental consent given? Results:  - The first sentences regarding wheeze are difficult to understand compared to fig 1. This may be clearer presented. - Results of FEV1 should be given as percent of predicted and the reference material given. Absolute lung function has little information in this age, especially as the predicted volumes may differ substantially depending on exact age, gender, height. Even more, FEV1/FVC ratio will be a better measure of obstructive lung function. - To analyse risk factor for absolute FEV1 including anthropometrics is not appropriate. It is obvious that the absolute value of FEV1 is associated with growth variables, also noted by the authors in the discussion. Therefore, these results has no value and may omitted, it is mandatory to use percent of predicted. - What is higher IgE ? Total IgE ? Total IgE has rather poor association with specific atopy. Discussion:  - Wheeze and asthma are not defined, and how the authors go from wheeze to asthma is not discussed. - If omitting several variables and research questions from the study, the discussion may be substantially rewritten focusing on the main research questions remaining and of primary value. This includes the relation wheezing and asthma, relation to atopy or not. - The limitations of spirometry at the age of six must be acknowledged. Some language editing needed OR with CI may be given as: example 1.27 (1.01, 1.35)
--	--

REVIEWER	Cara Bossley King's College Hospital, London
REVIEW RETURNED	10-Dec-2019

GENERAL COMMENTS	This is a large observational cohort study of the incidence of wheeze in young children in Havana, Cuba. The children were carefully followed up and had questionnaires, lung function and skin prick testing. This was a large study inclusive of over 1000 children and was well conducted. Methods Lung function 10% as positive for bronchodilator reversibility seems low - we would use a 12% cut off. Results 38% reported a wheeze - how was this checked - there could be recall bias or a misinterpretation of the term 'wheeze'. A large number had bronchodilator reversibility, but how many had this if you use a cut off of 12% which is often used as a cut off for bronchodilator reversibility? Risk factors for decreased FEV1 Municipality of residence was associated with FEV1. Was there any ethnic variation between municipalities which could account for this? Also, did you look at social class/income and how this impacted upon FEV1? Discussion The prevalence of self reported wheeze was high at 38% but this was on self reported data, and may include some patients who did not suffer from true wheeze. I agree that the fact that lung function was obtained in 82% of patients was excellent, and would congratulate you for this part of the study. I agree that the fact that 51% of homes contained a least one smoker is a worry and an important finding. The fact that you found that smoking was a risk factor for wheeze, which has been shown in other studies means that this is a significant healthcare concern, and one of the reasons I think this paper is important. The municipality of residence and the association with wheeze, could be looked into in more depth, maybe you could attempt to perform further air pollution or social screen analysis to explain this, or even add some further theories into the discussion.
---

VERSION 1 – AUTHOR RESPONSE

Reviewer 1 (Cara Bossley)

This is a large observational cohort study of the incidence of wheeze in young children in Havana, Cuba. The children were carefully followed up and had questionnaires, lung function and skin prick testing. This was a large study inclusive of over 1000 children and was well conducted.

Comment 1

10% as positive for bronchodilator reversibility seems low - we would use a 12% cut off.

Authors response

Thank you. We done this which resulted in changes to the abstract, results, discussion and Figure 2.

Comment 2

38% reported a wheeze - how was this checked - there could be recall bias or a misinterpretation of the term 'wheeze'.

Authors response

We used the methodology developed by the ISAAC consortium to explore international differences in asthma, and the information was provided by the parents/guardian. This has been used in Spanish speaking countries previously, and is the best available question/terminology available. This has been added to the methods along with a reference.

'This included a number of health and lifestyle questions that were answered by the parent or guardian and particular attention was paid to parental/guardian reported wheeze in the past 12 months using the methodology developed for the ISAAC epidemiological studies of asthma¹, use of asthma medication in the past 12 months and exposure to environmental tobacco smoke.'

Comment 3

A large number had bronchodilator reversibility, but how many had this if you use a cut off of 12% which is often used as a cut off for bronchodilator reversibility?

Authors response

We have changed this as suggested (see comment 1).

Comment 4

Municipality of residence was associated with FEV1. Was there any ethnic variation between municipalities which could account for this?

Authors response

The Cuban people are very heterogenous from a genetic/ethnic perspective, and there has been little inward migration to the island in recent decades. The Cuban authors consider that studies by ethnicity are not helpful as confidently labelling ethnic group is challenging. This is contrary to the UK where many ethnic groups may be first or second generation immigrants and can thus be given a label for an ethnic group with some confidence.

Comment 5

Also, did you look at social class/income and how this impacted upon FEV1?

Authors response

We collected some data on social class and income but decided not to use it as it was not likely to be helpful. This is because Cuba is a socialist country, where many people have a similar income. However, some people may get money from abroad or work in the black market. These people would be unlikely to disclose this information to our data collectors.

Comment 6

The prevalence of self reported wheeze was high at 38% but this was on self-reported data, and may include some patients who did not suffer from true wheeze.

Authors response

We used the self-reported definition of wheeze defined in the International Study of Asthma and Allergies in Childhood (ISAAC), as this is the commonly used accepted tool for large epidemiological studies. As asthma is a dynamic disease that will vary over 12 months, this is the best option that was available to us. We have added this limitation to the methods section.

'Collecting data on parental or self-reported wheeze using questionnaires is challenging, and although we used the optimal methodology from the ISAAC international studies of allergic disease ¹ it is possible that some of the cases of wheeze were not asthma.'

Comment 7

I agree that the fact that lung function was obtained in 82% of patients was excellent, and would congratulate you for this part of the study.

Authors response

Thank you. We appreciate your kind words and it is a tribute to the Cuban team who did the hard work.

Comment 8

I agree that the fact that 51% of homes contained a least one smoker is a worry and an important finding. The fact that you found that smoking was a risk factor for wheeze, which has been shown in other studies means that this is a significant healthcare concern, and one of the reasons I think this paper is important.

Authors response

We agree.

Comment 9

The municipality of residence and the association with wheeze, could be looked into in more depth, maybe you could attempt to perform further air pollution or social screen analysis to explain this, or even add some further theories into the discussion.

Authors response

We are currently doing a nested-case control study of objective measures of indoor air pollution in the study population to address this suggestion. We have added this comment to the discussion.

'We are exploring these hypotheses further by doing further objective measurements of particulate air pollution within the study population.'

Reviewer 2 (Knut Øymar)

Comment 1

This study seems to be asking a huge number of research questions which it is not primary designed for, no power analyses for this included, not a proper selection of variables and adjusted analyses. This is particularly for the high number of possible risk factors, of which the study provides no new information. I would suggest to concentrate on the main topic, the prevalence of symptoms, (possible lung function) and medication use. These are the main results of interest, providing that the authors can describe properly that the population is really unselected at the first inclusion.

Authors response

This is a cohort of children who were recruited at age 12-15 months and initially studied to try and identify risk factors for asthma in children of this age in Cuba ². We subsequently obtained further funding to look at these factors and also other risk factors for asthma found in the tropics such as infections. A lot of these exposures are novel with regard to asthma, and little is known about them, especially in developing countries in the tropics.

Many authors who collect data for cohort studies write a series of papers, testing one hypothesis at a time. However, it can be unclear what other hypotheses have been tested, found to be negative and are not taken forward to publication. Considering that we have already published some data from this cohort, and that we have been awarded a grant to test further hypotheses from the cohort that has also been approved by an Ethics Committee, we have put everything into one manuscript, which we believe to be an honest and transparent approach. Then the reader can look at the results with the confidence intervals and decide for themselves. We have emphasized this important point in the discussion.

'Like many cohorts, our study began as a cross-sectional population-based that aimed to explore causes of the high prevalence of asthma in Habana, Cuba. Further funding permitted this to become a prospective study, with a particular emphasis on infections found in the tropics. As a consequence we have tested a variety of analyses and present the results in their entirety. While we recognise that it is possible to generate a number of papers analysing different hypotheses from these data, this approach has the advantage of presenting a complete body of work that hopefully can inform researchers who are interested in this area.'

Comment 2

Measuring and analyzing lung function as absolute volumes is not appropriate, the percent of predicted and the used reference material should be given.

Authors' response

This is one of the first ever population-based studies of lung function in children aged 6 years old who live in a developing country. The definition and validation of lung function in adult populations is controversial with various adjustments for ethnic groups to generate predicted values. There are few reliable reference ranges for lung function in this age group, and these are all derived from developed, affluent countries, with very different life experiences, exposures and genetic constitution that that found in Cuba. Using a reference range to generate percent-predicted lung function values in this population would be flawed for this reason. Hence, we have decided to study lung function within the study-population adjusting for age in months (despite the fact that they are of similar ages) sex as a *priori* confounding factors. This analysis provides reliable outputs as the comparison is within a Cuban population. We have added this to the discussion.

'There are no validated lung function normal values in Cuban children aged 6 years old and as a consequence we were unable to generate reliable percent predicted values of the lung function in our study population.'

Comment 3

The authors justify that there is no selection bias from the first to the last follow up even though only 57% are followed the whole study. However, it must be better described the primary inclusion - are the included at health centers providing a selection of children with symptoms or randomly form all children in the municipality.

Authors' response

We are sorry for the lack of clarity. We have added the following to the methods.

'The study population is a cohort of 1956 children aged 12 to 15 months who were randomly selected from the general population from four municipalities across Havana in 2010 and 2011²⁻⁴.'

Comment 4

? Could even though parents' children with symptoms be more susceptible to accept?

Authors' response

There are many potential sources of bias that can occur in epidemiological studies. As can be seen, the original study population had a response rate of 96%. There was no difference in the proportion of children who wheezed in the first year of the study who attended in Year 6 or not which is the best test we can do to explore if children with symptoms were less likely to drop out of the cohort.

'The response rate of those who were eligible to participate initially was 96% ².'

'We initially recruited 1956 children to the cohort at the age of one year with a response rate of 96% eligible children, and approximately five years later were able to collect data on 1106 (57%) of these.'

'There was only a small difference in the prevalence of wheeze in the first year of life between those who provided data at the age of 6 years (46%) and those who did not (42%), so loss to follow-up is unlikely to constitute a major source of bias.'

Comment 5

Abstract:

The primary selection procedure could be better described, it is not easy to understand how the participants are included at first even if it is stated that it is a population-based study.

Authors' response

We have review this and clarified it as below.

'The study population is a cohort of 1956 children aged 12 to 15 months who were randomly selected from the general population from four municipalities across Havana in 2010 and 2011 ²⁻⁴.'

Comment 6

Based on the results the population seem selected.

Authors' response

We think that this comment is because of the very high prevalence of wheeze and use of steroids in Cuba. These are true data from a population-based study, not a population of children taken from an asthma clinic. This showed the scale of the problem of under-treated asthma in Cuba, and possibly other developing countries as well. This is a big public health concern for any country with similar results.

Comment 7

The abstract includes several different issues, it is not a full congruence between aims, methods, results and conclusions. If fewer research questions are included in the manuscript, the abstract may be rewritten.

Authors' response

Please see the response to comment 1. We could have written up a very clear, positive study report if we selectively chose a few interesting hypotheses, but this would not have been 100% transparent or honest. The analysis includes the research questions that were selected before the data were collected.

Comment 8

Methods:

The inclusion of patients may be better described, even though more described in previous studies. As main results are different prevalences, it is crucial to understand if there has been any selection. Lung function; a threshold of 200 ml is applicable for adults, not for children.

Authors' response

We have looked and the patient inclusion and made it clear that there is no selection and that this is a population-based cohort (text below). We used the ATS/ERS guidelines to provide a standard operating procedure for the lung function and are simply describing what we have done.

'The study population is a cohort of 1956 children aged 12 to 15 months who were randomly selected from the general population from four municipalities across Havana in 2010 and 2011²⁻⁴.'

Comment 9

At age six, a substantial proportion of children will not achieve acceptable spirometry data, here it seems that all children providing lung data are successful, making the results questionable. Even more, reversibility of 10% in this age as a measure of reversibility is low, and many children will probably achieve some improvement by chance or due to technical improvement after several measures.

Authors response

82% of our children studied provided a usable lung function measurement, which is consistent with the reviewers first statement, although it is not clear why he/she thinks this value is 100%. We have changed the definition of reversibility to 12% as mentioned in the response to reviewer 1.

Comment 10

A substantial numbers of unadjusted analyses (even though adjusted for age and sex) may not be appropriate. Fewer variables should have been carefully selected and included in multivariate analyses including sex and age, and considering interactions.

Authors response

As mentioned in our response to comment 1, in the interests of transparency we have tested all hypotheses that were considered important prior to data analysis. If we produced 3-4 papers looking at these some of these associations in isolation, there is a danger that we may be considered selective in our choice of hypotheses.

Comment 11

How are data regarding medications selected ? By questionnaire? A recall bias may be a problem, though perhaps than more probably an under-reporting.

Authors response

'This included a number of health and lifestyle questions that were answered by the parent or guardian and particular attention was paid to parental/guardian reported wheeze in the past 12 months using the methodology developed for the ISAAC epidemiological studies of asthma¹, use of asthma medication in the past 12 months and exposure to environmental tobacco smoke.'

We have also added the possibility of recall bias to the discussion.

'Similarly, parental reported asthma medication use may be prone to recall bias, although as acute asthma in children is an upsetting family event it is likely to be remembered accurately although the time period possibly less so.'

Comment 12

Ethics: were parental consent given?

Authors response

Yes. We have added this to the appropriate part of the methods.

'Ethics approval

The study was approved by Ethics Committees in the Instituto Nacional de Higiene, Epidemiología y Microbiología, Cuba and at the University of Nottingham, UK. This involved parental consent on the behalf of the child.'

Comment 13

The first sentences regarding wheeze are difficult to understand compared to fig 1. This may be clearer presented.

Authors response

Thank you. We see the point and have tried to make it clearer.

'Wheeze in the first year of life was reported in 514 (46%) current participants, while there was a prevalence of wheeze in the first year of life of 42% (358 children) for those who did not participate in the study at the age of six years (p=0.055).'

Comment 14

Results of FEV1 should be given as percent of predicted and the reference material given Absolute lung function has little information in this age, especially as the predicted volumes may differ substantially depending on exact age, gender, height. Even more, FEV1/FVC ratio will be a better measure of obstructive lung function. To analyse risk factor for absolute FEV1 including anthropometrics is not appropriate. It is obvious that the absolute value of FEV1 is associated with growth variables, also noted by the authors in the discussion. Therefore, these results has no value and may omitted, it is mandatory to use percent of predicted.

Authors response

This issue was discussed as a response to comment 2. In essence we are looking at factors that may modify lung function and asthma symptoms in a unique population of children who are all of a similar age, and who have a very high prevalence of asthma. As a consequence we have analysed differences within the Cuban study population which we consider to be the best approach available.

Comment 15

What is higher IgE ? Total IgE ? Total IgE has rather poor association with specific atopy.

Authors response

Thank you. We see the point. We have clarified the association between IgE and bronchodilatation.

'IgE at the age of one year was positively associated with a higher risk of bronchodilatation (+1.68%; 95%CI: +0.54 to +2.82), but not IgE at age of three years or six years.'

Comment 16

Discussion:

Wheeze and asthma are not defined, and how the authors go from wheeze to asthma is not discussed.

Authors response

The study was initiated as a consequence of concerns among clinician about asthma in Cuban children. Wheeze has been defined in the methods.

'This included a number of health and lifestyle questions that were answered by the parent or guardian and particular attention was paid to parental/guardian reported wheeze in the past 12 months using the methodology developed for the ISAAC epidemiological studies of asthma¹, use of asthma medication in the past 12 months and exposure to environmental tobacco smoke.'

The question of when wheeze becomes asthma is a clinical one, and hence susceptible to consulting bias. Hence we have reported wheeze throughout the methods. However, by the age of six years much wheeze is asthma, and so the discussion talks of 'asthma symptoms'. In terms of the public health significance of these data, they resonate with the concerns of the Cuban clinicians who initiated the study, and hence we do talk about the concerns about asthma cautiously. However, we also mention that not all wheeze is asthma to temper the discussion.

Although it is possible that children who have had wheeze subsequently received asthma treatment, this lack of association between asthma symptoms and objective measures of asthma has been described previously⁵.

Comment 17

If omitting several variables and research questions from the study, the discussion may be substantially rewritten focusing on the main research questions remaining and of primary value. This includes the relation wheezing and asthma, relation to atopy or not.

Authors response

Using the main research questions of primary value generates a potential risk of a *post hoc* cherry-picking the most 'interesting' results and is a very subjective approach to hypothesis generation. We are committed to delivering the hypotheses that were selected before the data were analysed, even if they are negative results. Anything else would not reflect well on our transparency and honesty.

Comment 18

The limitations of spirometry at the age of six must be acknowledged.

Authors response

We have done this.

'The measurement of lung function is a challenge in young children, yet measurements were obtained in 82% of eligible children.'

Comment 19

Some language editing needed

OR with CI may be given as: example 1.27 (1.01, 1.35)

Authors response

This is a stylistic issue which we can do if the BMJ Open journal requires it. The medical statistics courses in the UK teach defining the confidence intervals clearly so that there is no room for misunderstandings.

1. Committee TISoAaAiCIS. Worldwide variations in the prevalence of asthma symptoms: the International Study of Asthma and Allergies in Childhood (ISAAC) Steering Committee. *Eur Resp J* 1998;12:315-35.
2. Venero-Fernandez S, Suarez Medina R, Mora Faife E, et al. Risk factors for wheezing in infants born in Cuba. *Quarterly Journal Medicine* 2013;106:1023-29.
3. Suarez-Medina R, Venero-Fernandez S, Mora Faife E, et al. Risk factors for eczema in infants born in Cuba. *BMC Dermatology* 2014;14:6.
4. Fundora Hernández H, Suarez-Medina R, Venero Fernandez S, et al. What are the main environmental exposures associated with elevated IgE in Cuban infants? A population-based study. *Tropical Medicine and International Health* 2014;19:545-54.
5. MacKenney J, Oyarzun M, Diaz P, et al. Prevalence of asthma, atopy and bronchial hyperresponsiveness and their interrelation in a semi-rural area of Chile. *Int J Tuberculosis & Lung Dis* 2005;9:1288-93.

VERSION 2 – REVIEW

REVIEWER	Knut Øymar Stavanger University hospital
REVIEW RETURNED	23-Jan-2020

GENERAL COMMENTS	Thank you for the ability to review the revised version. The authors have replied to all questions and concerns. I acknowledge there are some issues to consider different in this setting then in a high-income setting. However, there are still some major and minor issues to consider. Major comments
---

	The authors reply generally that they have to report on the original design of the study, but that does not limit us to consider the quality of the scientific method and consequently the value of the results. Some strengths of the study are:  - The population based design with a high degree of inclusion primarily, now better described – and probably there is not an important selection for subsequent follow-up with high impact on the outcome related to numbers with wheeze and medication. - The lung function measures with reversibility (though also possible limitations as discussed below) In my opinion the main and important results from this study are the rate of wheezing / asthma reported, the low number of ICS users compared to systemic steroids, and perhaps also the rate of bronchoconstriction (with limitations discussed below). Analyses of risk factors and associated results may have limited value due to design and methods as discussed below. Lung function measures;  - 82 (83?)% successfully qualified for data, but with < 200 ml variation of FEV1. This value is for adults, and a variation of 200 ml will be too large for young children (mean FEV1 1,13 l) and do not secure that there is satisfactory low variation and satisfactory quality. If not strict measures for quality are used, any variation after bronchodilator could be due to training or variation. Measures according to guidelines for children would be needed to be able to rely more on the outcome of reversibility. - Absolute lung volumes are highly associated with height, and thereby also other anthropometric data, and with sex. Reporting associations between anthropometric data and absolute lung function as a result of the study has in my opinion no meaning, this is physiology. Even though there are no normal values available for Cuba, using available predicted values could be better. Simply adjusting for height and sex could also be a possibility. - Due to the above, other variables associated with lung function could be due to confounding. Statistics; the analyses are mainly a high number of unadjusted analyses, not considering interactions, confounding etc. Just reporting these associations may have some value, but limited because there is no adjusting for confounders (except for age and sex), and no plan for analyses a priori. Not all methods are described in this section of methods (chi2). A major result is the association of municipality with outcomes could also be due to confounding, in the analyses this is only checked for current smoking habits.
--	---

REVIEWER	Cara Bossley Consultant in Paediatric Respiratory medicine King's College Hospital Denmark Hill London UK
REVIEW RETURNED	15-Jan-2020

GENERAL COMMENTS	I am now happy with the revised version of the manuscript.
--

VERSION 2 – AUTHOR RESPONSE

Reviewer 1 (Cara Bossley)

I am now happy with the revised version of the manuscript.

Authors' response.

Thank you.

Reviewer 2 (Knut Øymar)

Thank you for the ability to review the revised version. The authors have replied to all questions and concerns. I acknowledge there are some issues to consider different in this setting than in a high-income setting. However, there are still some major and minor issues to consider.

Major comments (1)

The authors reply generally that they have to report on the original design of the study, but that does not limit us to consider the quality of the scientific method and consequently the value of the results.

Some strengths of the study are:

- The population based design with a high degree of inclusion primarily, now better described – and probably there is not an important selection for subsequent follow-up with high impact on the outcome related to numbers with wheeze and medication.
- The lung function measures with reversibility (though also possible limitations as discussed below)

In my opinion the main and important results from this study are the rate of wheezing / asthma reported, the low number of ICS users compared to systemic steroids, and perhaps also the rate of bronchoconstriction (with limitations discussed below). Analyses of risk factors and associated results may have limited value due to design and methods as discussed below.

Authors' response.

We agree that the population-based design, high use of systemic steroids and objective measures of lung function and bronchoconstriction are what make these data important from a public health perspective. We have presented the data as transparently as possible so that the reader can evaluate all the analyses of the data for themselves.

Comment 2

Lung function measures; 82 (83?)% successfully qualified for data, but with < 200 ml variation of FEV1. This value is for adults, and a variation of 200 ml will be too large for young children (mean FEV1 1,13 l) and do not secure that there is satisfactory low variation and satisfactory quality. If not

strict measures for quality are used, any variation after bronchodilator could be due to training or variation. Measures according to guidelines for children would be needed to be able to rely more on the outcome of reversibility.

Authors' response.

We understand the concern regarding the measurement of the variation in FEV₁ and how young children have small lungs that will impact on the reproducibility of the lung function measurements. We have acknowledged this issue in the discussion and will also bear this point in mind for future studies of this end point in this cohort.

'Finally, we used the 200ml as the definition of repeatability of FEV₁, which is a value generally used in adults. As a consequence, our population provided FEV₁ measures that will have a higher measurement error than observed in clinical patients. Nonetheless, these data of the peak FEV₁ value records have an epidemiological value as the methodology of data collection was standardised across the whole population. This is supported by the observation of the association between FEV₁ and height, which are both different aspects of somatic growth.'

Comment 3

Absolute lung volumes are highly associated with height, and thereby also other anthropometric data, and with sex. Reporting associations between anthropometric data and absolute lung function as a result of the study has in my opinion no meaning, this is physiology. Even though there are no normal values available for Cuba, using available predicted values could be better. Simply adjusting for height and sex could also be a possibility. Due to the above, other variables associated with lung function could be due to confounding.

Authors' response.

We have re-analysed the data of absolute lung function adding height into the regression model to ensure that all analyses are adjusted for height and are not hence a consequence of confounding by this variable which is associated with somatic growth (please see the track changes document for the precise changes). We have modified the statistical methods and results accordingly.

'As height was associated with FEV₁, all analyses of this outcome measure also adjusted for height to ensure that the analyses were not confounded by somatic growth.'

Table 3. Association of exposures with Forced Expiratory Volume in one second.

Total number = 903	No (% , sd)	FEV ₁ , ml (95% CI)
Prior exposures		
Any wheeze in 1 st year of life	416 (46)	-16 (-50 to +18)
Family history of asthma	510 (56)	-23 (-89 to +43)
Attendance at nursery	141 (16)	-48 (-149 to +53)

Paracetamol in 1 st year of life	215 (24)	+8 (-36 to +51)
Mean birth weight, N=, Kg (sd) N=901	3.32 (0.53)	+43 (-19 to +104)
Mean birth height, cm (sd)	50 (2)	+11 (+5 to +18)
Breastfeeding \geq 4months	503 (56)	+18 (-65 to +101)
Age = 1 year		
Mean log IgE, N=456	+3.44 (1.39)	-19 (-50 to +12)
Mean log eosinophils, N=451	-2.14 (1.04)	-13 (-56 to +30)
Age = 2 years		
Helicobacter stool +ve, N=589	29 (5)	-78 (-270 to +115)
Age = 3 years		
Mean log CRP, sd, N=614	-2.16 (2.71)	+2 (-9 to +12)
Log dengue IgG, N= 545	-0.43 (2.00)	+7 (-10 to +25)
Helicobacter stool +ve, N=775	51 (7)	-39 (-132 to +54)
Mean log eosinophils, N=652	-1.78 (1.26)	+22 (-5 to +49)
Toxoplasmosis serology +ve, N=606	362 (60)	+48 (-46 to +143)
Mean log IgE, N=614	+3.68 (1.57)	+8 (-21 to +37)
Medical diagnosis of dengue infection	78 (9)	-27 (-59 to +5)
Current exposures		
Male Sex	468 (52)	+36 (-23 to +96)
Mean age (months), (range)	74 (64 to 83)	+4 (-12 to +21)
Wheeze in past 12 months	353 (39)	-62 (-160 to +35)
Mean weight, Kg, N=902 (range)	23.5 (11 to 51)	+11 (+3 to +18)
Current height, cm, N=903 (range)	119 (90 to 175)	+8 (+2 to +14)
Mean arm circumference, cm N=901	18.4 (2.4)	+12 (+1 to +24)
Number of current smokers in house		
0	447 (49)	0
1	261 (29)	0 (-20 to +19)
\geq 2	195 (22)	-39 (-108 to +30)
Municipality of residence		

Arroyo Naranjo	371 (41)	0
Cerro	119 (13)	+74 (+31 to 117)
Habana del Este	247 (27)	+48 (+39 to +57)
La Lisa	166 (18)	-95 (-126 to -64)
		p<0.001
Current infection and blood assays		
Helicobacter stool antigen +ve, N=685	10 (1)	-10 (-228 to +207)
Toxoplasmosis +ve, N=688	437 (64)	+8 (-49 to +65)
Mean log dengue IgG serology, N=687	+1.63 (1.56)	-5 (-24 to +14)
Log CRP, N=686	-0.87 (1.38)	-9 (-37 to +20)
Mean log eosinophils, N=785	-1.61 (1.47)	-5 (-41 to +31)
Mean log IgE, N=688	+4.52 (0.97)	-12 (-87 to +63)
Toxocariasis +ve, N=667	57 (9)	-15 (-125 to +95)
Any atopy* (N=818)	52 (6)	-54 (-182 to +74)

adjusted for sex, age in months, current height and clustering by municipality

*defined as any allergen skin prick test > 3mm larger than saline control

FEV₁ = Forced Expiratory Volume in one second

CRP = C Reactive Protein, sd= standard deviation

Comment 4

Statistics; the analyses are mainly a high number of unadjusted analyses, not considering interactions, confounding etc. Just reporting these associations may have some value, but limited because there is no adjusting for confounders (except for age and sex), and no plan for analyses a priori. Not all methods are described in this section of methods (chi2). A major result is the association of municipality with outcomes could also be due to confounding, in the analyses this is only checked for current smoking habits.

Authors' response

We had the option of using this very rich dataset to write 4-5 papers on environmental, growth, blood parameters and infectious disease exposures on the outcomes of wheeze, lung function and bronchodilatation to salbutamol. However, this would exposed us to the possibility that some readers may have questioned our probity in selecting analyses of interest. Age and sex were obvious *a priori* confounding factors, and we have added height to the lung function analysis as stated above. However, we do not have the ability to adjust every possible association between exposure and

outcome measure presented for possible confounding and interactions, which is the trade-off of presenting all the data in one manuscript.

The association of lung disease in children with municipality is an important observation from the Cuban perspective, and current smoking habits was considered to be the most important environmental exposure that needed consideration based on prior knowledge of the Cuban context. We did not go fishing for other possible explanatory variables as we aim to drive the analysis from a public health perspective rather than simply putting variables into the model and seeing what happens. We have clarified this in the statistical analysis section of the methods and the discussion.

'Chi-squared tests were used to explore differences in categorical exposures for binary outcome measures.'

'While we have considered that sex and age in months are a priori confounding factors and adjusted for them in all analyses, we have refrained for searching through all the data to look for other possible confounding factors and possible interactions. The one exception to this is to explore the hypothesis that cigarette smoking may explain the differences observed between municipalities for wheeze, FEV₁ and bronchodilatation as this is an obvious question that was driven by Cuban public health concerns about the hazards of exposure to second-hand tobacco smoke.'

VERSION 3 – REVIEW

REVIEWER	Knut Øymar Pediatric Department, Stavanger University Hospital, Stavanger, Norway
REVIEW RETURNED	04-Feb-2020
GENERAL COMMENTS	Thank you for thorough considerations and revision and I have noe further comments.